# Bilirubin-Induced Neurological Damage: Current and Emerging iPSC-Derived Brain Organoid Models

**DOI:** 10.3390/cells11172647

**Published:** 2022-08-25

**Authors:** Abida Islam Pranty, Sara Shumka, James Adjaye

**Affiliations:** Institute for Stem Cell Research and Regenerative Medicine, Faculty of Medicine, Heinrich-Heine University, Moorenstrasse 5, 40225 Dusseldorf, Germany

**Keywords:** BIND, kernicterus, UCB, iPSCs, organoids

## Abstract

Bilirubin-induced neurological damage (BIND) has been a subject of studies for decades, yet the molecular mechanisms at the core of this damage remain largely unknown. Throughout the years, many in vivo chronic bilirubin encephalopathy models, such as the Gunn rat and transgenic mice, have further elucidated the molecular basis of bilirubin neurotoxicity as well as the correlations between high levels of unconjugated bilirubin (UCB) and brain damage. Regardless of being invaluable, these models cannot accurately recapitulate the human brain and liver system; therefore, establishing a physiologically recapitulating in vitro model has become a prerequisite to unveil the breadth of complexities that accompany the detrimental effects of UCB on the liver and developing human brain. Stem-cell-derived 3D brain organoid models offer a promising platform as they bear more resemblance to the human brain system compared to existing models. This review provides an explicit picture of the current state of the art, advancements, and challenges faced by the various models as well as the possibilities of using stem-cell-derived 3D organoids as an efficient tool to be included in research, drug screening, and therapeutic strategies for future clinical applications.

## 1. Introduction

Bilirubin is an endogenous toxin that results as a by-product of hemoglobin breakdown. It is often used to diagnose liver and blood diseases and has a complicated metabolism, which is significant in relation to various drug metabolism pathways [1]. Bilirubin is metabolized in the liver by the enzyme encoded by the uridine diphosphate glucuronosyltransferase 1A1 gene (UGT1A1), which conjugates bilirubin to glucuronic acid, making it water-soluble [2]. Being lipophilic, unconjugated bilirubin (UCB) cannot take part in the physiological elimination process and starts accumulating. Conjugation of bilirubin is required for increasing its solubility in plasma, thereby enhancing bilirubin elimination from the body. Furthermore, the high level of UCB can become dangerous and cause various complications.

When bilirubin levels in plasma or serum cross the laboratory reference range due to bilirubin metabolism irregularities, it is diagnosed as hyperbilirubinemia, which can be further categorized as conjugated or unconjugated hyperbilirubinemia (UHB) [3]. Clinical jaundice, for instance, which is caused by neonatal UHB, is a commonly occurring, transitional condition that affects about 85% of newborns in their first week of postnatal life [4,5,6]. UHB is a condition regulated by the albumin-bound UCB. As a consequence, there is enhanced UCB production, reduced conjugation and dysfunctional hepatic uptake [3]. On the other hand, at mildly elevated concentrations, bilirubin has a protective antioxidant-like effect on the body [7,8]. It can neutralize reactive oxygen species (ROS), prevent oxidative damage, and is even necessary for newborns when they face high concentrations of oxygen in the air for the first time [9,10,11,12]. It has been shown that UCB also possesses potent anti-oxidant properties, and modest hyperbilirubinemia may even have health benefits [1]. However, high levels of UCB can pose serious threats, such as severe brain injury, with the possibility of progressing into chronic bilirubin encephalopathy (also referred to as kernicterus) in one in every 100,000 cases, if not treated immediately [6,7]. In cases where it does develop into kernicterus, almost 70% of newborns die within the week and the other 30%, suffer irreversible brain damage [13].

Apart from being an adverse effect of spontaneous neonatal hyperbilirubinemia, BIND may also result from a genetic disorder known as Crigler–Najjar Syndrome (CNS). This life-threatening disorder is caused by the mutation in *UGT1A1*, which causes a complete or partial defect that prevents the liver from metabolizing bilirubin. This hinders bilirubin conjugation, causing UCB to accumulate in serum and eventually cross the blood–brain barrier, proceeding to deposit in the basal ganglia or cerebellum, thereby resulting in BIND [2,3]. Table 1 offers a comprehensive overview of various clinical indications related to hyperbilirubinemia and the respective targets in the brain.

BIND is not only temporarily disabling but also permanent, and it is usually accompanied by movement disorders as well as hearing loss [4,6]. The targeted damage to the central nervous system reflects the regional topography of bilirubin-induced neuropathology, involving the globus pallidus, subthalamic nucleus, metabolic sector of the hippocampus, hippocampal Cornu Ammonis (CA2) neurons, and Purkinje’s cells of the cerebellar cortex and the brainstem, as well as the oculomotor and ventral cochlear nuclei [14,15,16]. However, the key cellular mechanisms accounting for this well-defined regional topography of bilirubin sensitivity are still unclear. One probable reason could be the lack of efficient and suitable in vitro and in vivo models with consistent and comparable findings [12]. Advanced stem-cell-based studies offer great opportunity to establish 3D in vitro model systems to study neurological complications [17,18,19,20]. On that account, human-induced pluripotent stem cell (iPSC)-derived 2D neuronal cell cultures along with 3D brain organoids present convenient and efficient models to enable deciphering the molecular mechanisms underlying BIND. 

### 1.1. UGT1A1 

Glucuronidation is a conjugation reaction in which glucuronic acid, which is produced from the cofactor UDP-glucuronic acid, is covalently bound to a nucleophilic functional group on a substrate [28]. The *UGT1A1* gene, or uridine diphosphate glucuronosyltransferase 1A1 gene, is part of the UGT1 locus, which encodes the enzymes that glucoronidate a variety of substrates. This gene plays a crucial role in the glucuronidation pathway by converting bilirubin from an unconjugated (toxic) state to a conjugated (nontoxic) state [2]. Bilirubin is formed as a by-product of the heme catabolic pathway. After hemoglobin is broken down into heme, it is then transformed into biliverdin (BLV) and subsequently into bilirubin. *UGT1A1* particularly encodes the enzyme that has the ability to convert small lipophilic molecules such as bilirubin into hydrophilic (water-soluble) molecules that can be easily excreted [29,30]. During bilirubin glucuronidation, glucuronic acid is attached (conjugated) to bilirubin through a bilirubin-UDP-glucuronosyltransferase (B-UGT) enzyme-dependent reaction, as B-UGT1 is the only enzyme capable of glucoronating bilirubin [2]. The glucuronidation process takes place in the liver; therefore, liver cells are the primary source of the B-UGT1 enzyme. Thereafter, the water-soluble conjugated version of bilirubin is dissolved in bile and excreted from the body with solid waste. 

*UGT1A1* was initially cloned by Ritter et al. in 1991 and is located on chromosome 2q37 [30,31]. The UGT1 locus has 13 unique promoters and alternate first exons, followed by four common exons, designated 2, 3, 4 and 5. Before transcription, one of the first exons and its promoter are spliced to the four common exons. This results in 13 different UDP-glucuronosyltransferases being expressed; however, out of 13 possible genes that can be encoded, the only one responsible for bilirubin conjugation is the one containing the alternate exon A1 [2,30]. 

Reduced expression and partial or total impairment of the B-UGT1 enzyme is caused by mutations in the *UGT1A1* gene’s common or bilirubin-specific domains. This can result in inherited unconjugated bilirubinemia disorders, with the most common ones being Gilbert syndrome, Crigler–Najjar syndrome type I (CNS-I) and Crigler–Najjar syndrome type II (CNS-II), also known as Arias syndrome (Table 1) [4,29,30]. The nature of these mutations varies, resulting in phenotypes that range from moderate, in the case of Gilbert syndrome, to severe in CNS-I [4,29,32,33].

In 2000, Kadakol et al. tabulated more than 50 genetic lesions of *UGT1A1* that engender CNS-I and II and presented a correlation of structure to function of *UGT1A1* [29]. Building upon that research, almost a decade later, Canu et al. published an explicit list of Gilbert and CNS disease, causing mutations including more than 130 cases. Single-nucleotide changes were liable for around 70% of these alterations, whereas deletions, insertions, and polymorphisms attributed for the remaining 30% of alterations [32]. As *UGT1A1* is the crucial player in these diseases, being able to alter its expression in customizable in vitro models could help provide more insights into possible translational treatments.

### 1.2. Crigler–Najjar Syndrome Type I and II

The most serious form of inherited UHB is Crigler–Najjar syndrome Type-I (CNS-I) [34,35,36]. It is the outcome of the complete aberration of *UGT1A1*, a very rare autosomal recessive disease, only affecting one in a million individuals [37]. The aberration of *UGT1A1* leads to high bilirubin plasma levels and severe jaundice in neonates [2]. Increased bilirubin availability in plasma may result in bilirubin accumulation in the brain, turning into a life-threatening condition known as bilirubin encephalopathy. Crigler–Najjar Syndrome Type-II (CNS-II) and Gilbert syndrome are two milder versions of CNS-I, where *UGT1A1* is either partially deficient or altered, resulting in a less severe phenotypic manifestation [37]. 

To lower the plasma bilirubin level and prevent bilirubin encephalopathy, CNS patients rely on 10–12 h of intensive phototherapy treatment every day. Numerous dermatological disorders have been safely and successfully treated with phototherapy for over 40 years [38]. This treatment uses UV radiation to counteract the pathological changes that characterize inflammatory skin diseases through several mechanisms, such as induction of apoptosis, modification of the cytokine milieu, and immunosuppression. Phototherapy is so effective because through UV radiation, bilirubin is irreversibly photo-altered into lumirubin, a structural isomer that is more water-soluble, less dangerous and can be expelled with bile and urine [38,39]. However, the efficiency of the phototherapy can decrease depending on multiple factors, such as age, thickness of the skin, etc. Conversely, skin thickening is one of the effects obtained from the phototherapy itself, which later decreases the therapeutic efficiency. Additionally, extremely low-birth-weight newborns might face potential toxicity due to aggressive phototherapy [40]. A hemolytic process is indicated with the enhancement of total serum bilirubin level despite intensive phototherapy. Exchange transfusions have also been used to control hyperbilirubinemia at a hazardous level and lower the risk of kernicterus. However, phototherapy has greatly reduced the necessity and demand for exchange transfusion [6]. Another approach to control hyperbilirubinemia and prevent acute bilirubin encephalopathy is intravenous immune globulin therapy. Despite the mechanism being unclear, the immune globulin therapy seems to have biological activity against immune-mediated hemolytic diseases associated with the lowering effect of the immune globulin present on the total serum bilirubin level [6]. Pharmacological compounds may provide a direct protection to the neurons from bilirubin toxicity. CNS-II patients respond quite well to the pharmacological therapies, such as treatment with phenobarbital, whereas CNS-I patients do not. Bilirubin conjugation is increased by the activated phenobarbital enhancer module of the *UGT1A1* promoter sequence, thus resulting in enhancement in bilirubin clearance. On the other hand, heme oxygenase inhibitors, such as metallophyrins, can be employed to reduce bilirubin production [41]. Minocycline, which is a tetracycline antibiotic, has shown protective effects in Gunn rat pups against bilirubin-induced neurotoxicity, including neuromotor dysfunction, abnormalities in the auditory pathway and cerebellar hypoplasia [6,42]. Finally, liver transplantation remains the only effective treatment for this life-threatening disease (Table 2) [2,36,43].

## 2. Unravelling the Mechanisms Underlying BIND 

To increase our meagre knowledge of BIND, we must understand the pathophysiology underlying high bilirubin neurotoxicity at the molecular level. The brain is a highly specialized and compartmentalized organ with divergent cell populations consisting of neurons and glia, which comprises astrocytes, oligodendrocytes and microglia [48]. Therefore, the location, source and causal agents of BIND are the primary areas worth investigating, along with the cascade of molecular and cellular events that lead to severe damage.

Autopsy results of jaundiced neonates showed disperse yellow spots in the majority of brain areas, except the basal ganglia and medulla oblongata, while intense coloring was observed in those particular areas [49]. These observations indicate that UCB binds to specific types of neurons compared to others and has distinct sensitivities amongst neurons and glia [50]. Microscopic observations of jaundiced brain sections revealed the presence of bilirubin within neurons, neuronal processes and microglia; however, the contribution of individual neuronal cell types and cell-dependent sensitivity towards bilirubin toxicity are still not clarified [48]. In vitro studies have revealed the mechanisms associated with UCB neurotoxicity [48,50]. An increased impairment of cell function has been observed in astrocytes upon high UCB exposure, while neurons show higher susceptibility to cell death [50]. Astrocytes and microglia also seem to play key roles in activating oxidative stress and inflammatory responses. Investigation into intracellular processes of astrocyte and microglia showed that TNF-alpha and IL-1beta pathways as well as MAPK and NF- κB pathways play a key role in cytokine production and cytotoxicity upon UCB stimulation, resulting in UCB-induced neurotoxicity [5,51]. In vivo and in vitro data indicate oxidative stress to play a major role in cytotoxicity upon highly concentrated (toxic) UCB exposure, while increases in oxidative stress and cytotoxicity were observed in synaptic vesicles, tissue culture cells, and primary cell culture of neurons, astrocytes and oligodendrocytes [5,52,53]. 

The creation and elimination of bilirubin both result from a sequence of metabolic reactions; therefore, there are distinct ways of limiting the production and degradation of UCB [7]. The heme catabolic pathway primarily regulates bilirubin conjugation, as UCB is the consecutive end-product and UCB is endogenously produced by following this pathway within the majority of cells [54]. Briefly, heme is converted into BLV by heme oxygenase enzyme 1 and 2 (HMOX1, HMOX2), and then BLV reductase (BLVR) converts BLV into UCB. Both HMOX1 and HMOX2 reside in mitochondria, endoplasmic reticulum (ER) and caveolae (membrane micro-domains observed at the interface with the extracellular environment), which might have a correlation with BIND-induced neurocytotoxicity, as various molecular pathways become activated in the course of BIND neurotoxicity. These pathways include inflammation, mitochondrial damage and oxidative stress to ER [5,54,55]. The disturbances in mitochondria and ER usually lead to several additional sequelae, such as neuronal excito-toxicity (a complex process triggered by glutamate receptor activation resulting in dendrite degeneration and cell death), mitochondrial energy failure, increased intracellular calcium concentration and deoxyribonucleic acid (DNA) damage (Figure 1) [16,56,57]. All of these factors may subsequently contribute to neuronal death and bilirubin encephalopathy, leading to kernicterus [5,7,16]. 

During moderate to severe neonatal jaundice, pre-term newborns show an accelerated susceptibility to UCB toxic effects, which makes prematurity a significant abrasive factor for UCB encephalopathy [58,59]. The first week of postnatal life might be sensitive due to an increased chance of higher amounts of UCB availability in the circulation due to several factors. Consequently, the conjugation probability of UCB is suppressed and the unbound fraction of UCB (free bilirubin) increases [60]. The entry of UCB in the brain is restricted by the blood–brain barrier (BBB), as BBB is composed of tightly jointed microvascular endothelial cells, forming elaborate junctional complexes and providing unique properties by strictly regulating the ions, molecules and cell movement between blood and brain [60,61]. Lower UCB binding capacity and the higher UCB availability facilitate the entrance of free bilirubin by passive or facilitated diffusion into the brain, thus causing a condition of mild or severe hyperbilirubinemia. Further research is required to increase our meagre understanding of bilirubin entrance into the brain and the resulting cytotoxicity [59].

### 2.1. Bilirubin-Induced Oxidative Stress

Bilirubin plays a dual role depending on the physiological level of its unconjugated form. At very low levels, it acts beneficially as an antioxidant; however, after attaining a given threshold, it becomes toxic [5,62]. The neuroprotective role of bilirubin within a certain range of concentrations has been known for more than two decades to protect neurons from H_2_O_2_-induced toxicity [62]. Furthermore, the role of bilirubin as an anti-inflammatory agent and a scavenger of ROS have been intensively studied for a long time [54,63,64,65,66,67]. With the help of nicotinamide adenine dinucleotide phosphate (NADPH) oxidase, bilirubin prevents the generation of superoxides, inhibits ROS production and regulates redox homeostasis. This implies that at lower concentrations, bilirubin is potentially involved in several important cellular signaling pathways, such as cell proliferation, apoptosis, inflammation, and immune system upkeep. Moreover, bilirubin has also been proven to be a powerful signaling molecule that can help guard against a variety of disorders linked to elevated levels of oxidative stress [54,68,69].

On the other hand, bilirubin itself is the cause of oxidative stress. Increased oxidative stress activates transcription factor NF- κB and also increases phosphorylation of mitogen-activated protein kinases (MAPKs), therefore resulting in cytokine production and cell toxicity [70]. It is also clear that neurons are more susceptible to oxidative damage than other cell types in the brain such as astrocytes [71]. Bilirubin-induced DNA damage was found to be significantly increased in vitro, when neuronal and non-neuronal cells were exposed to 140 nM of free bilirubin. As potential adaptive responses to repair the damage, bilirubin therapy triggered primary DNA repair pathways through homologous recombination (HR) and nonhomologous end joining (NHEJ) [7]. These findings add to our understanding of the mechanisms underlying bilirubin toxicity and may have implications for newborns with severe hyperbilirubinemia because DNA damage and oxidative stress may be another significant element causing neuronal death and bilirubin encephalopathy. Studies in Gunn rats and UGT −/− mice have additionally shown high levels of lipid peroxidation by sulfadimethoxine-induced hyperbilirubinemia, as well the activation of key oxidative stress markers [70,72,73,74]. We anticipate further therapeutic discoveries concerning the role of bilirubin in diseases related to oxidative stress, as the breadth of all its biological functions have yet to be fully uncovered.

### 2.2. Effects of UCB on the Brain

Autopsies of hyperbilirubinemic brains have shown UCB to be localized within neurons and microglia, which results in the loss of neurons, demyelination, and gliosis (Figure 1). On the other hand, along with inducing oxidative stress in cortical neurons, UCB also disrupts the dynamics of the neuronal network in hippocampal neurons or in immature developing neurons, making these early-staged neurons more susceptible to UCB-induced injury [75]. In isolated cell cultures, UCB impairs neuronal arborization and induces the release of pro-inflammatory cytokines from microglia and astrocytes. However, cell-dependent sensitivity to UCB toxicity and the role of each neural cell type are not yet understood [5].

Clinical manifestations of hyperbilirubinemia indicate higher selectivity of bilirubin towards damaged brain regions, which particularly includes its preference for basal ganglia, cerebellum, brainstem nuclei, peripheral and central auditory pathways, and hippocampus [12,21]. The increased selectivity towards these injured brain areas has been well-known to closely correlate with the clinical signs of hyperbilirubinemia. However, it is the impairment of intracellular defense mechanisms in these areas, rather than the accumulation of UCB itself, that plays the primary role in brain damage [12]. As a result, bilirubin may disrupt developmental processes while incorporating multiple overlap and co-morbid neurodevelopmental disorders [21]. The damage to the basal ganglia and cerebellum correlates with movement disorders, athetosis (slow, involuntary, and writhing movements of the limbs, tongue, face, neck, and other muscle groups) and abnormal tone; the damage to the auditory brain nuclei and inferior colliculi is correlated to the auditory dysfunctions and hearing loss; and the damage to the brainstem and hippocampus correlates with the impaired oculomotor brainstem response and impairments in memory and learning (Table 1) [12].

Barateiro and his colleagues [76] used a kernicterus mouse model to display axonal damage as well as myelination deficits and glial activation in brain regions that usually accompany the neurological sequelae observed in severe hyperbilirubinemia such as the pons, medulla oblongata, and cerebellum. The observations from the study indicate the cerebellum as the most affected area, displaying greater myelination impairment and glia burden, as well as a loss of Purkinje cells and a reduced arborization of the remaining ones. The increase in astroglial and microglial reactivity possibly emerges as a response to myelination injury. It has also been hypothesized that excessive accumulation of total serum bilirubin (TSB) in the early neonatal period may promote the activation of the gene responsible for myelin basic protein (*MBP*). The increase in MBP seems to correlate with the inhibition or lack of myelin sheath formation. This may occur in response to inflammatory insults that affected the brain in the first place, leading to the production of ROS, or it may be a compensatory response to the lack of functional MBP due to the damage [76].

The Brites lab demonstrated that neuronal growth impairment and cell death caused by UCB is mediated by nitric oxide (NO) and glutamate, modulated by microglia, and prevented by glycoursodeoxycholic acid and interleukin-10 (IL-10) [77]. In another study, Falcao et al., created a model where astrocytes abrogated the well-known UCB-induced neurotoxic effects by preventing the loss of cell viability, dysfunction, and death by apoptosis, as well as the impairment of neuronal outgrowth [78]. UCB-induced alterations on neurogenesis, spinogenesis, neuritogenesis and axonal cytoskeleton dynamics indicate the relevance of UCB in synaptic plasticity abnormalities and the long-term neurodevelopmental disabilities, thereby making pre-term infants more vulnerable towards BIND [5]. Ultimately, the critical dual role of UCB in the brain raises questions, such as (1) which exact mechanisms and physiological switches lead to this beneficial–toxic threshold and (2) how can we regulate this duality of UCB to our advantage for future clinical applications? Having a BIND model as close to the clinical manifestation as possible may help to answer these questions by allowing us to investigate the cellular and pathophysiological mechanisms caused by UCB entry and its further effects in the brain.

### 2.3. Epigenetic Alterations Due to Bilirubin-Induced Neurotoxicity

Epigenetic studies have shown bilirubin neurotoxicity to affect vital regulatory mechanisms by significant modulation of gene expression [79]. Epigenetic processes involve DNA methylation, RNA methylation, histone post-translational modifications, and non-coding RNAs (ncRNAs), among which histone acetylation plays a vital role in gene modulation for several neuro-biological processes, including synaptic plasticity, brain development, differentiation, maintenance, and survival [80,81,82]. Apart from affecting cell fate and behavior, the acetylation/de-acetylation-mediated changes in gene expression induce excitotoxicity, oxidative stress, increased calcium load, inflammation, and apoptosis [83,84] (Figure 1). The observed induced mechanisms indicate a probable link between epigenetic impairment in neurodevelopmental processes and the hyperbilirubinemic phenotype [79]. Following these leads, Vianello et al. used developing and adult Gunn rats to track histone 3 lysine 14 acetylation (H3K14Ac) level in the cerebellum and observed age-dependent alteration of H3K14Ac in hyperbilirubinemic conditions. Gene ontology analysis of H3K14Ac-linked chromatin also revealed 45% of genes to be involved in CNS development. This finding suggests that epigenetic modulation during development and maturation of the brain structure is one of the causes of cerebellum hypoplasia in hyperbilirubinemic Gunn rats. On the other hand, histone acetylation plays a role in controlling oligodendrocyte differentiation and myelin production, and the down-regulation of myelin-associated glycoprotein (*Mag*) is one of the known repercussions of bilirubin-induced disturbances of oligodendrocyte maturation [81,85]. Studies have reported down-regulation of *Mag* in vitro along with other BIND models, including in pre-term infants [76,79]. This indicates that oligodendrocyte maturation and myelination can be affected by altered histone acetylation due to bilirubin-induced neurotoxicity, both in physiological CNS development and post-demyelinated repair processes. Remarks from these studies confirm that epigenetically impaired neurodevelopmental processes in hyperbilirubinemia may have a correlation in bilirubin neurotoxicity [79].

## 3. BIND and CNS Disease Models

Generating and studying model systems that closely recapitulate the main characteristics of BIND and severe UHB, is of high importance for developing effective clinical treatments and therapies to gain a better understanding of the pathophysiological mechanisms underlying this condition. Figure 2 illustrates a general overview of the most common in vivo and in vitro models of BIND and CNS (Figure 2A,B), as well as why having a CNS patient-derived iPSC model would be a better option (Figure 2C).

### 3.1. Animal Models

Animal models are often able to bridge the gap that in vitro models fail to recapitulate, as they much better resemble the disease features manifested in patients. Bortolussi and Muro rigorously reviewed animal models used to study bilirubin neurotoxicity and metabolism as well as the in vivo mechanisms of hyperbilirubinemia [13]. The most widely used amongst these models is the Gunn rat. This strain of Wistar rats spontaneously developed a one-base deletion of exon 4 in the *UGT1* locus, thereby creating an in-frame premature stop codon. Since this codon is translated into a truncated protein lacking the transmembrane domain, it results in the deficiency of all members of the UGT1A1 iso-enzymes. The complete deficiency of UGT1A1 enzymatic activity causes hyperbilirubinemia in the Gunn rat, making it the first hyperbilirubinemia animal model to mimic the CNS-I syndrome. This model has enabled scientists to gather a considerable amount of knowledge on bilirubin metabolism and toxicity in vivo [4,86]. 

Despite having a mild phenotype, Gunn rats display life-long non-hemolytic UHB, which is an important feature of human CNS-I. In order to develop acute central nervous system dysfunction and recapitulate hyperbilirubinemia more precisely, Gunn rats are often treated with hemolytic drugs or albumin–bilirubin displacers, such as sulphonamides or erythrocyte-lysing agents such as phenylhydrazine. An application of this method to induce hyperbilirubinemia is direct administration of sulfadimethoxine, a displacer of bilirubin from albumin binding sites. This increases the fraction of free bilirubin migrating towards lipophilic tissues such as the brain and is accompanied by a drop of systemic bilirubin [13]. If left untreated, homozygous Gunn rats display abnormalities in the cerebellum and hearing impairments just like the respective human CNS-I phenotype; however, unlike the patient-manifested features, these rats reach adulthood and are fertile.

The *UGT1A*-null mouse is another popular in vivo model, which presents a much more severe phenotype than the Gunn rat, with aggravated neurological damage and consecutive death [87].

Using genetic tools and technologies enable the creation of the mutation. Constitutive and conditional knockout, knock-in and transgenic strains of mice have been generated by manipulating the mouse genome and have allowed for the further exploration of key aspects of this disease. With the disruption of *UGT1* exon 4 by neomycin cassette, scientists were able to generate the first bioengineered mouse model of severe UHB. Mutant mice are a good model to study CNS-I, as they do not express *UGT1A1* and display neonatal hyperbilirubinemia. However, these mice die within 11 days after birth, which makes the model inconvenient for broad-spectrum investigations and reproducibility.

These invaluable animal models have provided an undeniable contribution in understanding the mechanisms underlying severe neonatal hyperbilirubinemia. Nevertheless, they still leave an open question regarding the mechanism and pathology in the human brain, which emphasizes the establishment of a human cell-derived model system to provide more insights into the molecular basis of the disease. 

### 3.2. In Vitro Models 

Cell types of different origins are also used to model several aspects of bilirubin toxicity and its main sequelae such as oxidative stress, ER stress and DNA damage. In vitro cultures are being applied extensively to study bilirubin neurotoxicity. These cultures mainly include immortalized cell lines such as human neuroblastoma cell lines, HeLa cells, Hepa 1c1c7 mouse hepatoma cells and human U87 astrocytoma cells, as well as primary cultures of rat and mouse neurons, astrocytes, microglia, oligodendrocytes, endothelial cells, and embryonic fibroblasts (Figure 2). Even though these 2D systems do not mimic the in vivo cell–cell interactions nor the morphological and physiological complexities of the whole tissue, they still display various properties of the in vivo situation. Exposing different types of cells to different concentrations of UCB is one of the key methods in exploring BIND [13].

Hippocampal neurons are the most frequently used cell type for testing the response of neuronal cells to bilirubin. When exposed to bilirubin, these cells exhibit a reduction in axons and dendritic processes, increased cell death, oxidation, and mitochondrial dysfunction, as well as overexpression of protection mechanisms. Moreover, newly differentiated neuronal cell types that are less differentiated display higher sensitivity compared to mature differentiated neurons. Similarly, oligodendrocytes also display high bilirubin susceptibility. Oligodendrocytes downregulate MBP production with the consequent impairment of myelin sheath formation and neuronal axonal function. 

Organotypic cultures are another form of ex vivo model that can be used to study bilirubin toxicity; however, there are limited studies exploiting these models, particularly using hippocampal slices. These models were able to demonstrate the impairment of synaptic plasticity due to bilirubin toxicity as well as the involvement of microglia in the UCB-induced neurotoxicity. Thus, 2D cultures have been employed as in vitro models for decades to study the cellular response in biochemical and biophysical directions and have contributed towards the significant advancement in understanding cell behavior and bioactivities [88]. Despite of being well accepted, it cannot be denied that the cell bioactivities and interactions in 2D cultures deviate remarkably compared to the in vivo responses. Considering the urge of having a model that more efficiently mimics in vivo conditions, 3D culture models such as spheroids or organoids have emerged as a potential platform to study different physiological and pathological processes. Employing another dimension around 2D cells with the extracellular matrix (ECM) markedly impacts the cellular fate with respect to proliferation, differentiation, mechano-response, and cell viability [88,89].

### 3.3. IPSCs and Organoids as Tools for Disease Modeling

Creating a model that properly recapitulates the molecular events underlying a specific neurological disorder is not an easy task. The majority of studies attempting to model BIND and CNS rely primarily on either mouse models, which poorly represent the human pathogenesis and phenotype, or post-mortem tissues, which usually only reflect the final stages of the disease [20,90]. Embryonic stem cells (ESCs) have indefinite self-renewal capacity and plasticity to differentiate into somatic cell types in the embryo, which makes this cell type valuable for studying the mechanisms involved in specialized cells and organ development. ESCs offer a great opportunity for regenerative medicine by generating specialized cells based on different degenerative diseases and replace those with the damaged tissues [91]. However, derivation and application of ESCs remain ethically controversial, as the derivation process involves the use of human inner-cell-mass cells isolated from blastocysts [92,93]. Moreover, it is not always convenient to obtain samples, and the reproducibility of results is affected. These concerns can be side-lined by using human-induced pluripotent stem cells (hiPSCs), as these cells hold great promise for increasing our fundamental understanding of human biology during early development and pave the way for future regeneration therapies and personalized medicine. Recent advancements in gene editing technologies such as clustered regularly inter-spaced short palindromic repeats (CRISPR) have also made it possible to introduce genetic variants, for example through inducible gene knockout, thus opening new doors for in vitro disease modeling [94,95,96,97,98,99].

Further expansions in iPSC research have increasingly revealed the multifaceted use of these cells in modeling various diseases in vitro [17,18,19,20,100]. With the development of iPSCs, researchers have been able to replicate many diseases, including Parkinson’s disease, Nijmegen Breakage Syndrome, and Alzheimer’s disease, all by generating different types of cells that mimic the in vivo environment very closely [19,101,102,103,104,105,106,107,108,109]. For instance, deriving iPSCs from patients with genetic-based neurological conditions and differentiating them into neurons opens up more possibilities to closely observe the pathological mechanisms underlying the disease in vitro [106,110,111,112].

Shinya Yamanaka and Kazutoshi Takahashi introduced four defined factors; OCT3/4, SOX2, c-MYC, and KLF4, and established the pioneering protocol of generating iPS cells by reprogramming adult human fibroblasts [113,114]. Afterwards, Junying Yu et al. demonstrated another efficient combination of factors with OCT4, SOX2, NANOG, and LIN28 for reprogramming human somatic cells, which specifically exclude c-MYC [92]. Moreover, numerous studies have emerged in recent years, demonstrating the successful generation of iPSCs from different human somatic cells using integrating (retrovirus, lentivirus) and non-integrating (adenovirus, sendai virus, pSin plasmid, episomal plasmids, minicircle DNA) delivery systems [113,115,116,117,118,119]. With the other somatic cell (e.g., blood, urine cells)-derived iPSCs, it has been possible to avoid the invasive approach of skin biopsy, yet some methylation profile differences are still present between iPSCs and ESCs [120,121,122,123,124]. Nevertheless, iPSCs are ethically approved and considered identical regarding cell morphology, proliferation, and differentiation capacity, which also make it possible to generate large quantities of neuronal cultures for disease modeling, drug screening and therapy [19,92,125,126,127]. Additionally, iPSCs enable studying patient-specific disease conditions by reprogramming the cells obtained directly from the patient and therefore increase the scope to attain customized medication and therapy. 

The iPSC-derived 2D monolayer model is the classical approach for obtaining specific neural cell types to enable the investigation of cellular and molecular mechanism associated with healthy and disease states. Neural stem cells (NSCs) or neural progenitor cells (NPCs) have a self-renewing capacity and can differentiate into the neuronal lineage, resulting in multiple types of brain cells during mammalian developmental and adult stage (fetal to postnatal, through adulthood) [128,129,130]. However, NSCs show heterogeneity and high regional specificity in adults, while the newly differentiated neurons derived from the primary progenitors migrate and intermingle with specific brain regions [130,131]. The type of generated neurons is determined by the neuroepithelial origin of NSCs, which is linked to NSC localization and developmental timing regions [131]. There are various established protocols for generating NSCs derived from iPSCs (Figure 3). Adherent iPSCs are used to generate embryoid bodies (EBs) and these are then with specific growth factors such as epidermal growth factor (EGF), fibroblast growth factor 2 (FGF-2) along with B27 (without retinoic acid) and N2 in the medium to achieve neural rosettes. Afterwards, these neural rosettes can be re-plated in a monolayer culture to obtain NSCs [132,133]. On the other hand, neural rosettes can also be generated without EB formation by using ESC and iPSC colonies, which are detached and then treated with EGF and FGF-2 to grow as cell aggregates. These cell aggregates have the potential to form neural rosettes and are able to differentiate into a range of both central and peripheral neural lineages [132,134,135,136,137]. However, the NSCs obtained from neural rosettes may provide a heterogeneous and inconsistent proportion of differentiated cells, which can be avoided by deriving pure cultures of specific types of brain cells from iPSCs with specific inductors [138,139,140,141]. Overall, neural rosette formation and differentiating into specific cell types can be employed as a potent in vitro system to study human neurological diseases by uncovering molecular pathways. Nonetheless, some major limitations include distinction among different iPSC lines, batch-to-batch variability, and growth of rosettes in an irregular and non-coordinated manner. Even though it is possible to characterize and measure the quality of individual rosettes using different assays to some extent, the understanding of the dynamics from monolayer cells to a developed rosette is presently limited [137,142]. Conversely, generating 2D monolayered homogenous neuronal cultures by directed differentiations are financially and technically feasible, along with high-resolution cell morphology and great reproducibility. Guided differentiation to specific neuronal subtypes holds the potential for cell therapy or personalized medicine to treat neurodegenerative diseases [143,144]. However, the non-identical cellular age of the cells and the differences in differentiation, culture and maintenance procedure may also affect the comparability of the results [145].

The self-organizing capacity of hiPSCs to form whole tissues of various organ systems have evolved as a great advancement from 2D to 3D in vitro models [146,147,148]. In vivo methods provide complex and three-dimensional spatial arrangement to the cells, where circulating molecules, neighboring cell and the extracellular matrix are surrounding them [149]. Mono-layered mono- or co-culture systems lack this in vivo physiological relevance, which has a vital effect on cellular and physiological responses. In this regard, a three-dimensional system offers more physiological resemblance with respect to structural complexity. hiPSC-derived three-dimensional brain organoids recapitulate the key aspects of neurodevelopment along with reflecting some function of the system [150]. Three-dimensional organoids contain highly divergent cell types and subtypes, providing complex architecture and interplays with spatial organization. Being an intact tissue with spatial organization, organoid models offer the opportunity to observe the dynamic growth and development of the system over time [19]. Genetic mutations affect cell type, cell behavior, their interactions, neuronal network, and components of the various neurodevelopmental and physiological processes. iPSC-derived brain organoids afford studying these genetic mutations and multi-faceted brain diseases [108,148,151,152,153]. Moreover, being cultured in vitro, organoids provide easy accessibility genetically and for live assays [154]. 

Generally, cerebral organoids are composed of functioning neurons, astrocytes, oligodendrocytes and to some extent microglia [148]. Lancaster et al. established the protocol to generate self-patterned cerebral organoids containing forebrain, midbrain, hindbrain, and choroid plexus identity (non-directed differentiation) [148,155]. They used iPSCs to form EBs, which were gradually directed towards the neuroectodermal lineage, and then maintained these neuroectodermal tissues with extracellular matrix support in a spinning bioreactor to provide nutrition and a three-dimensional environment. With this approach, neural identity can be obtained in 8–10 days, resulting in defined brain regions by 20–30 days of culture, and the organoids can be cultivated for longer period to study later stages of neurodevelopment [154]. As the non-directed protocol relies on the cells’ differentiation and self-organization capacity without providing any inductive signals, it is considered as intrinsic and a non-manipulated system [156]. Cerebral organoid research has significantly expanded within the last decade with the introduction of more complexity and specificity [157]. Relying on small molecules, individual brain-region-specific organoids can be generated by growth-factor-based manipulation, which consequently determine the cellular identities such as cerebral organoids with choroid plexus, hippocampus, retina and striatum [158,159,160,161]. The generation of organoids by co-culturing different cells is another advanced and innovative approach to enhance the model complexity and to investigate the cell–cell and cell–matrix interplays in a 3D environment during human brain development and disease [162,163,164]. The co-culture systems in organoid technology reveals the mechanism of stem cell interactions, which might be useful in regenerative medicine study [165]. For example, iPSC-derived microglia (cells or assembloids) can be integrated into the midbrain organoids (generated from iPSCs), and then neurodegenerative and neuroinflammatory diseases can be investigated using this model. In contrast, microglia differentiation medium lacks neurotrophic factors, which are required for promoting dopaminergic neuron differentiation, resulting in lower numbers of dopaminergic neurons [166]. Consequently, the incompatibility between the small molecules in the medium might interact and affect the quantity and quality of existing cell types. Therefore, it is challenging to compose a perfect culture medium for co-culture organoids, as these comprise different cell types, where individual cell types require distinct medium compositions [165]. Additionally, a major drawback includes different proliferation rates of the co-cultured cells in the system, which might affect the maturity of the organoids and limit the long-term culture. On the other hand, release of paracrine factors by one cell type might affect the other cells either positively or negatively in this system [167,168]. Since both directed and non-directed organoid generation processes have benefits and drawbacks, the application of each should be determined according to the purpose of the study (Figure 3).

Considering various incorporated features in cerebral organoids, these models are closer to the developing human brain and mimic the neural environment much better compared to other in vitro models. This facilitates the understanding of disease pathology, drug mechanism and customized medication [108,153,169,170,171]. Cerebral organoids represent a higher degree of maturation and developmental dynamics mimicking the early second trimester of the fetal brain tissue; nevertheless, the accurate human brain equivalent age of the organoids still remains an unanswered question [172]. Furthermore, the organoid model shows high batch-to-batch variability in common with other iPSC-based models and requires sophisticated methods [19]. During slow development of the organoids, a tissue-degenerated necrotic core tends to form in the center due to the lack of optimal diffusion of nutrients and metabolites. Although the culture medium is oxygenated in the bioreactors, this is not enough to support culture for a longer period [148]. Despite the limitations, iPSC-derived brain organoids are a promising tool for 3D in vitro model systems, as they display functions and circuitry comparable to the human brain [148]. Even though they do not fully recapitulate the complexities of the human brain, they can still be a valuable study tool, as they are composed of distinct neural cell types important for the central nervous system [173]. Furthermore, human brain organoids have revealed useful insights into human brain development and successfully helped to model a variety of neurological disorders such as microcephaly, Timothy syndrome, and Nijmegen Breakage Syndrome, as well as brain tumors such as gliomas [108,148,174,175,176,177,178]. All of these models offer a solid platform for future of brain organoids as a valid tool for studying neurological disorders affecting the human brain [179].

UHB is an ailment observed in the first postnatal week, which can lead to acute or chronic UCB encephalopathy. The neonates show vulnerability towards UCB and have an increased risk associated with particular conditions, such as premature birth, sepsis, and hypoxia. Pre-term and low-birth-weight infants are even more vulnerable towards BIND due to neurodevelopmental immaturity, when sepsis or infection is incorporated [5,180,181]. Since brain organoids recapitulate key aspects of neurodevelopment and reflect certain functions of the system, they can therefore be exposed to UCB for modeling BIND. Both the immature and mature stage of cerebral organoids can be exposed to UCB for shorter (4–5 h) and longer (72 h–several days) periods to mimic the acute and chronic effect of UCB in the CNS. As UCB is a lipophilic compound, it should be able to penetrate the organoids. Additionally, iPSCs derived from CNS-I patients can be used to generate brain organoids, which will model the disease more precisely due to defective *UGT1A1*, and these organoids can be exposed to UCB to mimic the hyperbilirubinemic condition in the CNS [112]. As autopsy revealed the presence of UCB in neurons, astrocytes, neuronal process and so on, divergent cell types containing organoids will help us to understand the pathophysiology of BIND. Neurons are known to be more susceptible to UCB than astrocytes and generally demonstrate a higher level of ROS, protein oxidation and lipid peroxidation upon UCB exposure [71,182]. On the other hand, astrocytes cause morphological changes in mitochondria and ER when affected by high concentrations of UCB, leading to oxidative stress and cell death [183,184]. Furthermore, high levels of ROS produced by neurons upon UCB exposure results in oxidative stress in microglia [185]. Overall, UCB exposure affects the redox status of neurons and glial cells and induce inflammation with increased ROS, thus resulting in cell death in the CNS [71,182]. As cerebral organoids are composed of neuronal cell types and subtypes with some functionality and network complexity, it is possible to recapitulate the altered redox status induced by UCB toxicity and consecutive inflammatory responses using this model. However, it should be noted that bilirubin might also be considered as a neuroprotective compound when the concentration is below 100 nM [5,62]. In this regard, a kill curve should be performed to identify the suitable concentration for UCB, which can be used to mimic the hyperbilirubinemic condition in the CNS.

Furthermore, co-morbidity along with the degree and timing of UHB can affect an infant to develop one or multiple defects from the BIND spectrum. Therefore, studying the independent correlation of UHB with each neurodevelopmental disorder individually is not sufficient. Appropriate statistical analyses and power can be applied to evaluate possible co-morbidities of multiple neurodevelopmental disorders within the BIND spectrum, which may help us to define the association of each neurodevelopmental disorder with bilirubin-induced neurotoxicity [21,186,187]. Some of the parameters of bilirubin-induced neurotoxicity measurement include assessment of oxidative stress, DNA and RNA damage, post-transcriptional modifications, bilirubin accumulation in the brain and transporters, ER stress, inflammation and autophagy, which are also possible to study in the cerebral organoid model [13,70]. For instance, after UCB exposure to cerebral organoids, oxidative stress or impaired redox status can be monitored by glutathione (GSH) and oxidized glutathione (GSSG) measurements, where a lower ratio of [GSH]/[GSSG] indicates an increased oxidative state [70]. From transcriptome analysis of the treated organoids, bilirubin-induced ER responses can be observed by altered gene expression and regulation of ER stress-related genes (e.g., CHOP, ATF3, FAS) [74]. Gene ontology analysis can also reveal the connection between ER and inflammatory responses through distinct but relevant pathways (e.g., activation of p-ERK, NF-κB pathways) [70,188]. Moreover, assessment of pro-inflammatory mediators such as IL-6, IL-8, TNF-α, IL-1β by cytokine array or ELISA, can help uncover bilirubin-mediated inflammation [189,190]. UCB induced increased oxidative stress and ER stress, and neurodegeneration-mediated inflammation leads to apoptotic cell death, which can be detected by deoxynucleotidyl transferase-mediated deoxyuridine nick-end labeling (TUNEL) assay [70].

## 4. Concluding Remarks

To date, there are very few effective treatment options for CNS-I. To lower plasma bilirubin levels and prevent bilirubin encephalopathy, patients undergo daily phototherapy treatments, which inevitably become less effective as the patients age. Exchange transfusion is also sometimes used as an emergency treatment for neonates to rapidly lower serum bilirubin concentrations; however, this approach has been associated with serious complications, such as thrombocytopenia, portal vein thrombosis, necrotizing enterocolitis, and sepsis [44]. Liver transplantation remains the only effective treatment for this life-threatening disease, even though it does not reverse or alleviate pre-existing neurological damage [2,36,43,44,45].

Severe neonatal jaundice and hyperbilirubinemia remain a cause of devastating neurological damage in infants. Although this occurrence is rare, it can be completely avoided if the neonates receive treatment on time and the medical professionals prevent early discharge [12]. Currently, there is a clear gap in the knowledge we possess on the molecular mechanisms underlying this neurological damage. Therefore, creating a model of BIND based on genetically inherited disorders of the *UGT1A1* gene, such as CNS Type I and II, can help us further understand these mechanisms. 

Possible therapeutic approaches include anti-inflammatory-based medicines, gene manipulation and albumin infusion. Some other promising approaches include the modulation of nuclear receptors, cytochromes or BLVR activity, to control bilirubin production or to stimulate alternative bilirubin-disposal pathways. However, further research is needed before these techniques can be applied clinically. To shed light on human biology and health, a thorough understanding of the molecular pathways leading to bilirubin neurotoxicity is critical. 

Other potential therapies include hepatocyte transplantation, during which about 5–15% of the liver is replaced by transplanted hepatocytes, as well as gene therapy. Injections of naked plasmid DNA and adeno-associated virus gene therapies are currently being investigated, as preclinical models have been quite promising [44]. The constant hope with new emerging iPSC-derived 3D brain organoid models is that they can help shed light onto developing more effective ways of handling BIND and inherited unconjugated bilirubinemia disorders in the near future. Eventually, this model will enhance our understanding of the etiology underlying BIND and its pathology in the human CNS. This knowledge will aid in the development of drugs and future clinical applications.

## Figures and Tables

**Figure 1 cells-11-02647-f001:**
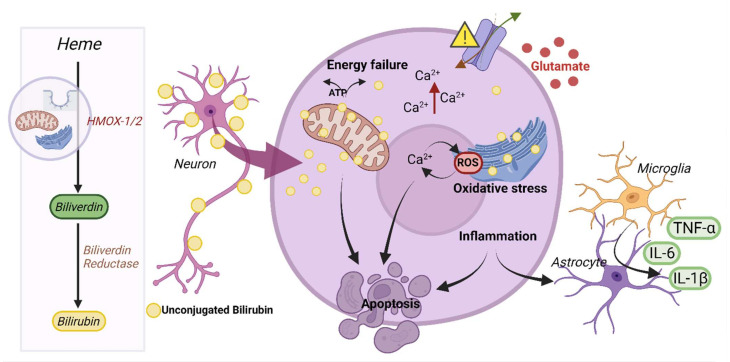
Schematic of the mechanisms involved in BIND induced neurotoxicity. **Left**: Metabolic pathway leading to unconjugated bilirubin (UCB) production. Heme is converted into biliverdin by heme oxygenase (HMOX-1 and 2), located in mitochondria, endoplasmic reticulum and caveolae. Biliverdin is then converted into unconjugated bilirubin by biliverdin reductase. **Right**: Neurons are depicted in this scheme to represent the toxic effects of UCB in brain cells. Neurons are known to be the most affected cell type in UCB toxicity, which involves multiple pathways leading to distinct toxic events, including disruption of the mitochondrial energetic breakdown, ionic imbalance, extracellular accumulation of glutamate, release of inflammatory cytokines by glial cells (here depicted as microglia and astrocytes), as well as increase in reactive oxygen species (ROS) production and oxidative stress. This UCB-induced cytotoxicity can result in apoptosis (the different cell type sizes are not depicted to scale, but rather schematically to simplify the view of the mechanisms) (Created with BioRender.com, accessed on 23 August 2022).

**Figure 2 cells-11-02647-f002:**
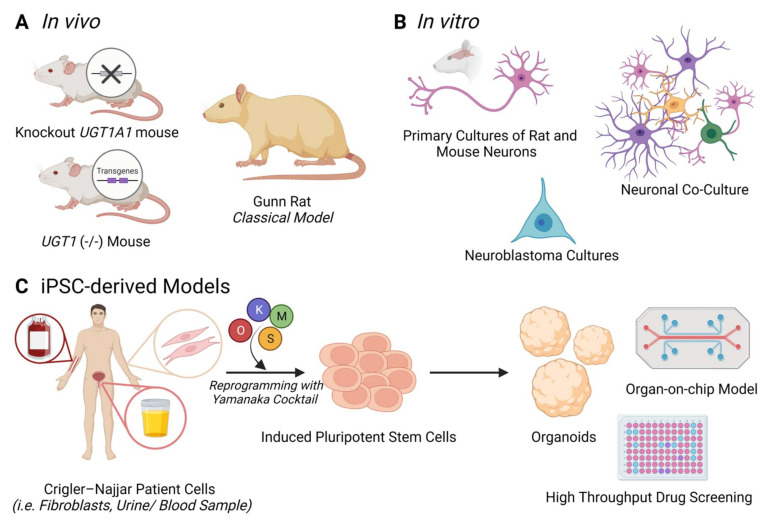
Overview of current models of BIND and Crigler–Najjar Syndrome. (**A**) In vivo models include the classical Gunn rat model, as well as knockout and transgenic mouse lines; (**B**) Some examples of in vitro models are primary cultures, mixed neuronal co-cultures as well as neuroblastoma cultures; (**C**) iPSCs can be generated by reprogramming of distinct types of human cells, such as fibroblasts, blood and urine-derived cells. The cells can be reprogrammed using the Yamanaka transcription factors (OCT4, SOX2, KLF4 and cMYC). A Crigler–Najjar Syndrome patient-derived iPSC model will enable future personalized medicine applications such as organoid cultures, organ-on-chip models and can be used for high throughput drug screenings specific to the patient’s needs (Created with BioRender.com, accessed on 21 August 2022).

**Figure 3 cells-11-02647-f003:**
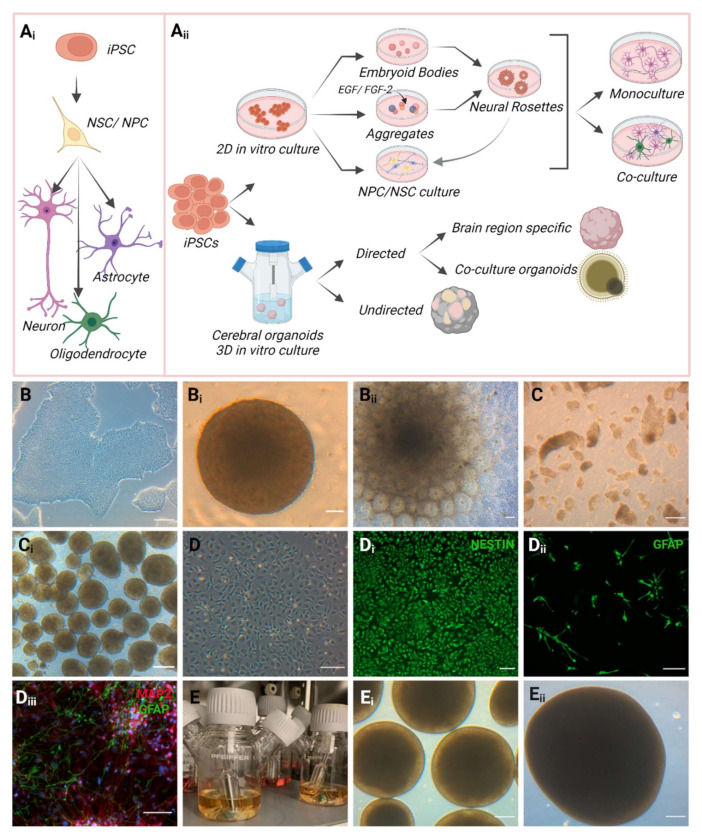
An overview of iPSC-derived 2D and 3D in vitro model generation with examples. Various protocols are available for generating iPSC-derived 2D and 3D in vitro models. (**Ai**) Schematic of iPSCs differentiation into neuronal cells. iPSC-derived neural progenitor cells (NPCs) have the capacity to differentiate into mature neuronal cells such as neurons, astrocytes, and oligodendrocytes. (**Aii**) NPCs can be obtained from 2D mono-layered iPSCs cultured as embryoid bodies (EBs) or cell aggregation in suspension via neural rosettes formation, or by direct induction from iPSCs to NPCs in 2D. NPCs generated in both approaches can be used for mono- or co-culture of distinct types of neuronal cells. Three-dimensional organoid cultures can be generated in a directed or non-directed manner, depending on their application purpose. (**B**) CNS-I patient-derived iPSCs 2D monolayer culture. (**Bi**) EBs generated from CNS-I patient-derived iPSCs. (**Bii**) Neural rosettes formation by replating EBs in 2D. (**C**) iPSCs are dissociated from the 2D culture to generate aggregates. (**Ci**) Cell aggregates generate spheroids in a shaking incubator. (**D**) Monolayered NPC culture. Immunofluorescence staining shows. (**Di**) Nestin-positive NPC cultures. (**Dii**) GFAP positive astrocytes. (**Diii**) MAP2-positive neurons (red) and GFAP-positive astrocytes (green) co-culture. (**E**) iPSC-derived 3D organoid culture in spinner flask. (**Ei**,**Eii**) Organoids at different time points of culturing. (Scales in 100 μm, bright-field and immunofluorescent staining images are taken from unpublished work in our lab) (Created with BioRender.com, accessed on 21 August 2022).

**Table 1 cells-11-02647-t001:** Overview of bilirubin-related diseases and clinical manifestations in the brain.

Clinical Indication	Brain Target	Clinical Symptoms	Reference
Bilirubin-induced cerebral cortex injury	Cortical neurons Astrocytes Oligodendrocytes	Reduction in neurite extension and dendritic and axonal arborization Increased cell death by apoptosis Cognitive disorders	[5,21,22]
Basal ganglia injury	Subthalamus Globus pallidus Striatum	Attention deficit hyperactivity disorder (ADHD) Specific learning disability (SLD) Cognitive and behavioral symptoms	[21,23,24]
Bilirubin-induced cerebellar injury	Cerebellum	Oxidative stress Endoplasmic reticulum (ER) stress Autism spectrum disorder (ASD) ADHD	[21]
Bilirubin-induced hippocampal injury	Dendrites and axons of hippocampus	Adverse synaptic plasticity Specific learning disabilities	[5,21,25]
Bilirubin-induced auditory nervous system injury	Brainstem auditory structure	Language disorders	[26,27]
Crigler–Najjar Syndrome Type I	Entire brain Particularly: basal ganglia cerebellum brainstem nuclei peripheral and central auditory pathway hippocampus	Mild to severe jaundice Kernicterus	[3]
Crigler–Najjar Syndrome Type II	Entire brain Particularly: basal ganglia cerebellum brainstem nuclei peripheral and central auditory pathway hippocampus	Mild jaundice Kernicterus (rarely)	[3]

**Table 2 cells-11-02647-t002:** Existing treatments and therapies for CNS.

Treatment	Advantages	Disadvantages	Reference
Phototherapy	Non-Invasive Relatively easy to administer Inexpensive	Time consuming/exhausting for the patient (10–12 h/day) Less effective as patients age Thickens patients’ skin which makes therapy less efficient	[6,44]
Exchange transfusion	Rapid treatment (in emergency cases lifesaving)	Thrombocytopenia Portal vein thrombosis Necrotizing enterocolitis Sepsis	[44]
Intravenous immune globulin therapy	Removes need for exchange transfusions	Fever Allergic reactions Rebound hemolysis Fluid overload	[6,41]
Liver transplantation	Most effective	Does not reverse or alleviate pre-existing neurological damage	[2,44,45]
Phenobarbital	Increased bilirubin clearance	Not applicable for CNS-I patients	[6]
Metallophyrins	Reduce bilirubin production	Photosensitivity Iron deficiency Affect hematopoiesis (the formation of blood cellular components)	[41,46,47]
Minocycline	Protective effects against neuromotor dysfunction, abnormalities in auditory pathway and cerebellar hypoplasia	Unsafe for newborns Affect bone and dentition development	[6]

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
