# Peer review of "Bilirubin-Induced Neurological Damage: Current and Emerging iPSC-Derived Brain Organoid Models"

_cells, 2022, doi:10.3390/cells11172647_

Round 1

Reviewer 1 Report

Dear authors,

the review is interesting but some improvements have to be done.

Firstly, you have to improve the English quality.

Secondly, you have to add some figure regarding the unravelling mechanism of BIND to clarify better the mechanistic part. Furthermore, you have to add some epigenetic section due to there are some articles about this. 

Consider adding some new references from the last 5 years.

Consider improving the rest of the schemes.

Consider adding some tables regarding the treatments.

Best regards,

Author Response

Dear Reviewer,

Thank you very much for your comments. We have answered all the comments addressed by you. We are convinced that your comments and feedback have helped to improve the quality of our manuscript.

Reviewer 2 Report

In this review, the authors aim to review models to study the mechanisms of bilirubin-induced neurological damage (BIND) and the use of current and emerging iPSCs-derived brain organoid models. 

However, they hardly deliver. First, most of the review is concerned with the mechanisms that lead to unconjugated hyperbilirubinemia. Only from page 7 on, a description follows on the mechanisms of BIND. Secondly, only one page is devoted to current iPSCs-based models, and only a single small paragraph mentions iPSCs-based brain organoid models, but fails to discuss this subject in any depth.  For example, what are the possibilities and (dis)advantages of developing brain organoids by mixing neuronal cell types each derived from cell-lineage specific iPSC-differentation versus unguided differentation from iPSC-derived NPCs? When to expose the brain organoids with unconjugated bilirubin, throughout development of the organoid, or at/from a particular time of organoid development? What exactly can you investigate in the organoid model that you cannot do in iPSC-derived cells in mono-or co-culture? What is the readout parameter for BIND using these brain organoids. This review therefore adds little to reviews for example by Amin et al (2019) and Bortolussi & Muro (2020).

What is the rationale to so prominently deal with  the detrimental effects of UCB on the liver function (abstract, table 1, page 5), and to expand on iPSC-derived hepatocytes (page 11)? Do UCBs-induced liver damage indeed result in the generation, or lack of removal, of toxic agents beyond UCB, that may be involved in BIND or may aggravate BIND?  

Minor:

Throughout: please give credits to the original studies, and not to the reviews (f.e. Amin et al 2019; Bortolussi & Muro 2020) that only mention these studies. 

Abstract: it is not customary to include references in the abstract.

p. 1: The sentence "The liver is able to..." needs correction.

p. 1: Please rephrase the sentence "Unconjugated bilirubin (UCB) is tightly...".

Table 1: Please explain the abbreviations.

p. 5: "Bilirubin is formed as a by-.......this small molecule breaks down heme." Does bilirubin indeed break down heme?

p. 5, the paragraph starting with "UGT1A1 was initially cloned...": "13 different ugt-ases being expressed", but further on there are only 9 UGT1A1 protein products; "the remaining nine 5'-exons", remaining from what?; with bilirubin being the enzyme's preferred..", but you have 9 different enzyme products from this gene, which of these 9 is meant?

p. 6: Chapter on UGT family members expressed in brain: Do all the different UGT family members glucuronate bilirubin? If not, then there is no added value of this chapter with the exception of 1A1. If yes, when does a UGT1A1 mutation have such a deleterious effect and do the other UGT family members not compensate?

p.7: "...limit the production and degradation of UCB." Do you mean that the described mechanisms involving mitochondrial and ER perturbations are because of degradation of bilirubin by these organelles, or that the functionality of these organelles is disturbed by the presence of excess UCB?

p. 7 last sentence: what is the controversy between being an antioxidant and being a scavenger of ROS?

p.8: "..higher selectivity of bilirubin towards damaged brain regions." Do the authors mean that brain regions must first be damaged by other reasons to preferentially accumulate bilirubin, or that there is a vicious cycle of bilirubin-induced brain damage followed by attraction of even more bilirubin to that brain region?

p. 9 first sentence: please explain how activation of the gene responsible for MBP leads to accelerated loss of MBP and lack of myelin sheath formation.

Figure 1A: Why mention HeLa cultures in particular where all other non-neuronal cell lines could be used to investigate general cellular toxicity?

p.11: "...for instance, cancer cells cannot be practically characterized and modelled in 2D culture": what is the relevance for this review?

p. 11: when discussing neuronal cell lineages from patient-iPSCs, one should also discuss the advantages and disadvantages of cell lineages derived from hESCs in which selected genes are altered by gene modification. 

p.12: Please rephrase the sentence "Anti-inflammatory based.... as potential therapeutic interventions": administration of drugs to engineered mice?  

Author Response

Dear Reviewer,

Thank you very much for your comments. We have answered all the comments addressed by you. We are convinced that your comments and feedback have helped us to improve the quality of our manuscript.
